# Experimental and Numerical Study on the Packing Densification of Metal Powder with Gaussian Distribution

**Huadong Yang [1,2,\*]**, **Shiguang Li [1]**, **Zhen Li [1]** and **Fengchao Ji [1]**

[1] Department of Mechanical Engineering, North China Electric Power University, Baoding 071003, China; miaomiaodaddy@163.com (S.L.); wllizhen@163.com (Z.L.); jfc15735182916@163.com (F.J.)

[2] Science and Technology on Plasma Dynamics Laboratory, Air Force Engineering University, Xi'an 710038, China

[\*] Correspondence: yanghd@ncepu.edu.cn; Tel.: +86-312-752-5433

**Abstract:** In the additive manufacturing of metal materials, powder bed fusion 3D laser printing is the most widely used processing method. The density of the packed bed is another important parameter that can affect the part quality; however, it is the least understood parameter and needs further study. Aiming at addressing the problem of the powder packing density in the powder tank before powder spreading, which is neglected in the existing research, a combination of numerical simulation and experimental research was used to analyze the powder particle size distribution, powder stiffness coefficient, and vibration condition. Considering the van der Waals forces between the powders, a discrete element model suitable for fine metal powders for 3D printing is proposed. At the same time, a mathematical model that takes into account the vibration state is proposed, and the factors affecting the density of the powder were analyzed. A self-designed and manufactured three-dimensional vibration test rig was used to conduct physical experiments on spherical metal powders with approximately Gaussian distributions to obtain the maximum densities. The results obtained by the numerical simulation analysis method proposed in this paper are in good agreement with the experimental results. The influence of the amplitude and vibration frequency on the powder packing density is the same; that is, it increases with an increase in amplitude or frequency, and then decreases with a further increase in amplitude or frequency after reaching the maximum. It is unreasonable to discuss the packing densification only relying on the vibration intensity. Therefore, it is necessary to combine the amplitude and frequency to analyze the factors that affect the packing density of powders.

**Keywords:** discrete element method; packing density; Gaussian distribution; metal powder

## 1. Introduction

Additive manufacturing (AM) is a layer-by-layer processing method. Powder bed fusion is an important method for manufacturing industrial-grade components, including selective laser melting (SLM), selective laser sintering (SLS) [1,2] and selective electron beam melting (SEBM) [3]. Because the powder bed method sinters a layer of powder to make the shape, the characteristics of the powder bed are very important for the quality of the printed part.

As is well known, the quality of the final part of additive manufacturing is affected by many influencing factors, such as powder quality, process parameters, gas flow distribution, and powder spreading quality. Researchers have performed a lot of work on the influence of process parameters on part quality. Recent studies [4,5] have found that the quality of the powder bed not only affects the density of the parts but also affects the interaction process between the laser and the metal powder, which in turn affects the formation of the microstructure and the performance of the parts. For the

quality of the powder bed, its influencing factors include the quality of the powder, the performance of the powder spreading system, and the parameters of the powder spreading. In addition, the initial powder packing state obviously also affects the spreading and distribution of the powder on the powder bed, which will affect the quality of the final parts, especially for powders with original defects. Therefore, it is urgent to the explore how to improve the initial powder packing performance in the powder cylinder. Additionally, A. Averardi [6] points out that the effects of the laser power, scanning speed, and powder bed layer thickness on parameters such as the porosity in the printed part, surface roughness, and microstructure have been studied. The density of the packed bed is another parameter of critical importance as outlined by many studies but is among the least understood parameters in this field [7,8]. In the current research, the impact of the packing of particles in the powder cylinder on the quality of the final parts is ignored. Therefore, this article focuses on this unresolved problem.

Metal powders are usually fed by a powder spreading or powder feeding system. For the powder spreading method, a straight blade travels in parallel with the substrate to spread a thin powder layer, and the laser beam scans the powder in the powder bed to selectively sinter the solid powder material to form a layer of the part. The final properties and accuracy of the manufactured component are highly dependent on the quality of the single powder layers [9–13]. The packing density, surface uniformity, and effective thermal/mechanical properties are important factors that affect the packing quality of the powder layer. For example, in SLM, a smoother and homogeneous powder layer makes the powder melting more stable and the final quality of the manufactured component better. Moreover, it can be found that the higher the packing density, the better the performance of the final parts. The quality of the thin layer of powder is related to the process parameters, including the powder size distribution, powder spreading speed, layer thickness, and scraper geometry [14,15]. In this paper, we focus on the packing density of the powder bed in SLM, which is shown in Figure 1. From this figure, it can be observed that the characteristics of the powder bed are related to the powder delivery system. Thus, it is important to analyze the packing density of the powder delivery system.

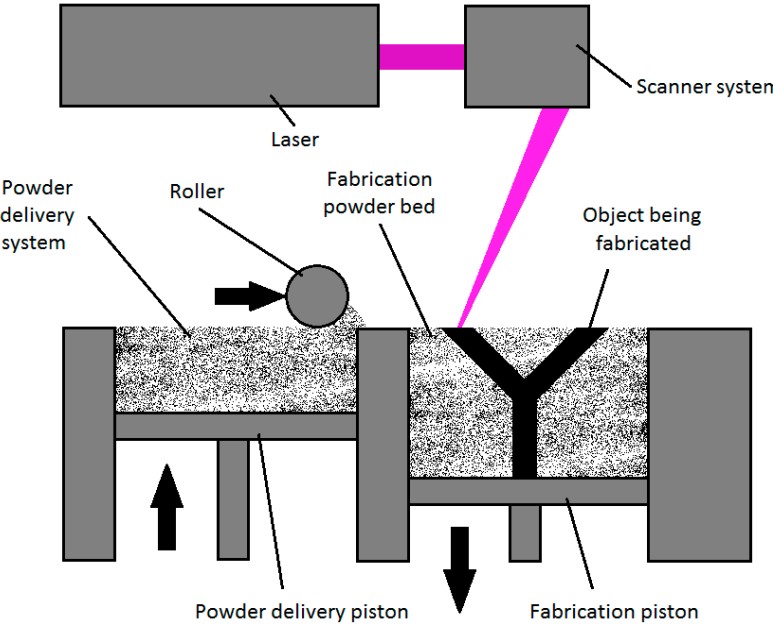

**Figure 1.** The principle of the selective laser melting (SLM) process.

The challenge in the powder bed is to eliminate the porosity in the printed part; therefore, it is important to understand the characteristics of the packed bed because improving the packing of the powder bed can reduce the defects and improve the heat transfer characteristics [16]. In the powder-delivering system, the powders are stored in the cylinder before being delivered by the scraper

blades. In the current research, the packing density of powders in the powder cylinder is ignored. The powder size distributions, cylinder sizes, and stiffness coefficients will affect the packing density of the powders in the cylinder, which will affect the packing quality of the powder layer.

In order to discuss the packing density of powders in a designated container, physical experiments and numerical simulations are usually applied. Li C. [17] conducted experimental studies on the packing density of mono-size and binary spherical particles under vibration conditions, and found that the maximum packing densities under one-dimensional and two-dimensional vibration conditions were 0.8294 and 0.8809. Most scientific research focuses on the analysis of the spherical particle packing structure, but between different spherical particles, the friction coefficient, the material stiffness, the particle size distribution, the damping coefficient, and other factors will affect the particle packing porosity. The packing of different particles in various containers under laboratory conditions and the detection of the internal structure of the particle were experimentally discussed by C.T. [18,19], so as to obtain the microscopic dynamic process for the particles. The packing and flow of particles in containers [20–22] and the influence of mechanical vibration on the particle packing in drum containers [23,24] have all been explored and studied by researchers through experimental methods. The process of particle packing generally goes through random loosening, random close packing, and orderly packing. Under the conditions of different vibrations, corresponding physical experimental verification and quantitative characterization are lacking. Therefore, the packing density of ternary or even more multi-size spherical particles needs further study.

In addition to experimental research, numerical simulation is also a very effective and less resource-consuming research method. For the numerical simulation of particle packing [25], Monte Carlo (MC) and the discrete element method (DEM) are widely applied. The MC method uses random numbers in the process of particle generation and movement. It can only provide a simple description of how many particles exist on each plane, and cannot accurately provide dynamic information of the particles during the packing process. The discrete element method uses dynamic relaxation, Newton's second law, and time-step iteration to solve the velocity and displacement of each particle, and is particularly suitable for solving nonlinear problems. At the same time, by changing the particle and boundary analysis model, contact mechanics model, and parameters, it is also possible to analyze the effect of the contact of different particles with different boundaries and its influence on the movement of particles. To sum up, it is not only suitable for the physical model involved in this topic to derive the dynamic characteristics of internal particles in the vibration state, but is also more conducive to the analysis of the boundary effect during the vibration process.

In order to analyze the packing density in a powder delivery system, a combination of numerical simulation and experimental research was used to study the effects of the powder stiffness, powder particle size distribution, and vibration state on the powder packing density. In this paper, through experiments and numerical simulations, the packing density of spherical powder particles with Gaussian distributions is explored. Specifically, a self-designed and manufactured three-dimensional vibration test rig was used to conduct physical experiments on spherical metal powders with approximately Gaussian distributions to obtain the maximum density. Considering the van der Waals forces between the powders, a discrete element model suitable for fine metal powders for 3D printing is proposed. At the same time, a mathematical model that takes into account the vibration state is proposed, and the factors affecting the density of the powder were analyzed. Finally, the experimental results were compared with the numerical simulation results in order to realize the high-density packing of spherical powder particles with Gaussian particle size distributions, and obtain the corresponding optimal parameters, which lays the foundation for further research on the density of spherical powder particles and guidance for industrial applications.

## 2. Test Rig

In order to accurately analyze the packing process for metal powder and the influence of vibration on the packing density, a three-dimensional vibration experiment test rig capable of realizing X, Y,

and Z axes perpendicular to each other was designed. The experimental test rig can realize not only simultaneous vibration in three directions, but also simultaneous vibration in any two directions and independent vibration in any single direction.

### 2.1. Configuration of Test Rig

Based on stability considerations, an I-shaped steel structure was selected for the base of the test rig to ensure the stability and flatness of the base. A two-dimensional front view of the experimental test rig is shown in Figure 2. The components mainly include a base (1), support column (2), shock-absorbing spring (3), bottom plate (4), third table from the top (5), upper second table (6), top table (7), sliding block (8), sliding rail (9), rib board (10), middle column (11), X,Y direction 1.5–2 vibration motor (12), and Z-direction 5–4 vibration motor (13). Other auxiliary devices and equipment include frequency converters, air switches, and organic glass containers.

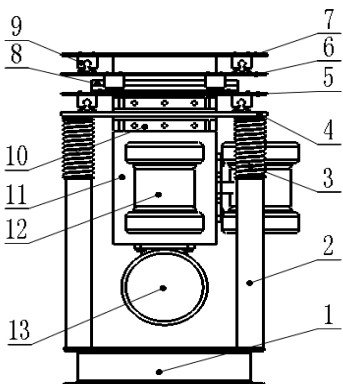

**Figure 2.** Schematic diagram of three-dimensional shaking table.

An image of the test rig is shown in Figure 3. During the experiment, amplitude control could be achieved by controlling the built-in eccentric of the vibration motor, and the control of the vibration frequency was achieved by adjusting the frequency of the inverter. The controlled variable method was used to measure the packing density of metal powder particles under different combinations of amplitude and frequency. The best matching parameters could be obtained. The vibration displacement is defined as in Equation (1).

$$S = A\sin(2\pi f t + \varphi) \tag{1}$$

where $S$ is the displacement (mm), $A$ is the amplitude (mm), $f$ is the vibration frequency (Hz), $t$ is the vibration time (s), and $\varphi$ is the primary phase. In the actual experimental process, the amplitude, vibration frequency, and vibration time were experimental parameters that directly affected the experimental densification results for particles. The speed $n$ of the vibration motor is proportional to the frequency $f$; the relationship is defined as in Equation (2).

$$n = \frac{N_e f}{60 f_e} \tag{2}$$

where $n$ represents the actual speed of the motor (r/s), $N_e$ is the rated speed of the motor (r/min), $f$ refers to the actual output frequency of the motor (Hz), and $f_e$ represents the rated output frequency of the motor (Hz).

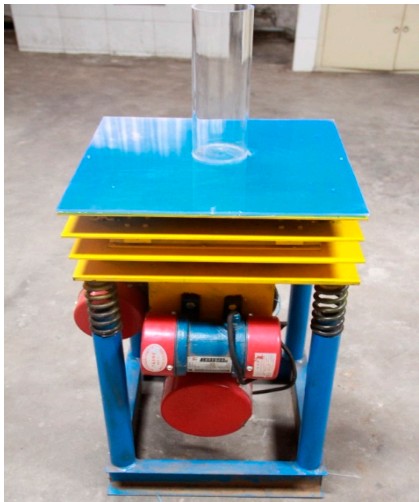

**Figure 3.** Image of three-dimensional vibration test rig.

*2.2. Materials in This Experiment*

In AM, powder feedstock materials are typically fabricated via water atomization, gas atomization, and plasma rotating electrode-comminuting process (PREP) [25–28]. The generated powder particles are of micron scale, the degree of sphericity is relatively high, and the particle size distribution generally shows a Gaussian distribution.

The Gaussian distribution density function of the particle size is as follows.

$$P\left(d_p\right) = \frac{1}{\sqrt{2\pi}} \times exp\left[-\frac{\left(d_p - \overline{d_P}\right)^2}{2\sigma^2}\right] \tag{3}$$

In the formula, $\sigma$ and $\overline{d_P}$ are two characteristic values of the normal distribution, $\overline{d_P}$ is the average particle size, and $\sigma$ is the standard deviation, and its value determines the dispersion of the particle size.

$$\sigma = \left[\frac{\sum n_i\left(d_{p_i} - \overline{d_P}\right)^2}{N}\right]^{\frac{1}{2}} \tag{4}$$

where $d_{p_i}$ represents the size of particles $i$, $n_i$ is the number of powder particles within the specified size range, and $N$ represents the total number of particles.

It is known that there is the following relationship in the normal distribution.

$$\overline{d_P} = \overline{d_L} = d_{50} = d_d \tag{5}$$

In the formula, $\overline{d_L}$ is the arithmetic average diameter, which is the ratio of the sum of all particle diameters to the total number of particles (also called the length average diameter). $d_{50}$ is the median diameter, which is the cumulative frequency of the particle size distribution equal to 50% of the particles' size. $d_d$ is the common diameter, which is the particle size corresponding to the maximum probability density $p$ value in the frequency density distribution curve.

In production practice, the particle size of the powder particles produced follows an asymmetric normal distribution, and the particle size deviates from the direction of larger particles. The particle size coordinates are replaced by logarithmic coordinates, and the distribution is similar to the symmetric

curve of a normal distribution. It is said that the particle size conforms to a lognormal distribution, and its function form is as follows.

$$q(d_p) = \frac{1}{\sqrt{2\pi} \times \ln \sigma_g} \times exp\left[-\frac{(\ln d_p - \ln d_g)^2}{2(\ln \sigma_g)^2}\right] \tag{6}$$

In the formula, $\sigma_g$ and $d_g$ are the two characteristic numbers of a lognormal distribution, $d_g$ is the geometric mean diameter ($d_g = d_{50}$), and $\sigma_g$ is the geometric standard deviation.

$$ln\sigma_g = \left[\frac{\sum n_i(\ln d_{pi} - \ln \overline{d_p})^2}{N}\right]^2 \tag{7}$$

In the actual sedimentation process for the particles, because smaller particles are prone to suspension or collision, there is a tendency for large particles to rapidly coagulate downwards, which further balances the internal particle distribution of the particles after sedimentation. Therefore, this paper mainly studies the packing density of particles with a normal particle size distribution.

The material used in this experiment was iron powder with an average particle size of 30 μm. An experimental container with a height of 300 mm was selected as the organic glass container because it has many advantages such as easy processing, small error, easy measurement and observation, and low cost.

### 2.3. Experimental Process Design

The packing density of powder particles in two-dimensional vibration is higher than that in a single-vibration condition. Preliminary experiments were carried out for two-dimensional vibration in horizontal and vertical directions, and the duration of each vibration experiment was roughly 30 s. At the same time, the output frequency of the vibration motor was adjusted through the frequency converter under different amplitudes, and then, the best experimental matching parameters were obtained. During this process, the amplitude and frequency in both horizontal directions were the same.

Before the experiment, the position of the built-in eccentric block in the 1.5–2 vibration motor and 5–4 vibration motor was firstly adjusted to determine the magnitude of the vibration amplitude, and the output frequency of the vibration motor was adjusted by the frequency converter. Secondly, the particle container was fixed to be vibrated. After loading the material, the height of the material in the container was measured. Finally, the vertical edges of each layer of the vibrating table were aligned to ensure that the vibration started from the geometric center of the plate surface, making the working process stable and reliable.

During the experiment, the displacement sensor was used to measure the vibration of three tables in the horizontal directions and the vertical direction when the vibration stabilized.

### 2.4. The Calculation of Packing Density

In order to reduce the influence of external factors on the experimental results, the particles and containers needed to be dried and cleaned before the experiment. Then, a certain number of particles were poured into the container. Before starting the vibration test rig, the height of the particles in the container was firstly measured by measuring 10 points evenly distributed around the circumference to improve the measurement accuracy.

The packing density can be obtained with the following equation.

$$\rho = \frac{V_p}{V_c} = \frac{m_p/\rho_p}{(\pi D^2/4) \times H} \tag{8}$$

where $V_p$ and $V_c$ are the volumes of the particles and container, $m_p$ is the quality of the particles poured into the container, $\rho_p$ is the density of the particles, $D$ is the diameter of the container, and $H$ is the height of the particles in the container. Then, the height of the particles vibrated by test rig with different vibration frequencies (f = 5, 10, 15, 20, 25, 30, and 35 Hz) was remeasured; thus, the packing density could be computed using the above equation.

## 3. DEM Simulation

### 3.1. The Model for Fine Powders

The general solution process for the discrete element method includes the following steps. (1) Discretize the solution space into a discrete element array and use reasonable connecting elements to connect two adjacent elements according to the actual problem. (2) The relative displacement between elements is the basic variable. From the relationship between the force and relative displacement, the normal and tangent forces between the two elements can be obtained. (3) The force between the element and other elements in all directions and the external forces caused by the action of other physical fields on the element are calculated according to Newton's second law of motion. The acceleration of the element can be obtained. (4) The forces are time-integrated to obtain the velocity and displacement of the element. In this way, physical quantities such as the velocity, acceleration, angular velocity, linear displacement, and rotation angle of all units at any time are obtained.

The physical characteristics of a single powder are treated as an independent unit in DEM simulation. The interaction of powders is not a purely elastic interaction. The friction of the powders and the relative movement of the powder contact cause the loss of kinetic energy and dissipate, thereby driving the overall transformation of the powders to a static state. Therefore, this method is very suitable for studying the packing densification of powders. The DEM method simulates the interaction force between the mass elements as the elastic force and damping force, which are calculated by mechanism models such as elasticity, damping, and friction slip. When two powders come into contact, the interaction in the normal direction can be simplified as a spring damper, and the interaction in the tangential direction can be simplified as a spring damper and a sliding friction device. When the directional friction force is greater than the maximum static friction force, the sliding friction device works.

In the simulation, a powder has two types of motion, which are shown in Figure 4, namely, translation and rotation. These two motions can be expressed by Newton's second law as follows.

$$m_i \frac{dv_i}{dt} = \sum_j \left( F_{ij}^n + F_{ij}^t \right) + m_i g \tag{9}$$

$$I_i \frac{d\omega_i}{dt} = \sum_j \left( R_i \times F_{ij}^t - \mu_r R_i \left| F_{ij}^n \right| \hat{\omega}_i \right) \tag{10}$$

where $m_i$, $v_i$, $\omega_i$, and $I_i$ indicate the mass, translational velocity, angular velocity, and moment of inertia of the powder $i$. $R_i$ is the vector pointing from the center of the powder to the contact point, and the length of the vector is equal to the radius of the powder. $\mu_r$ represents the rolling friction coefficient and can, respectively, represent the normal contact force and tangential contact force between powders. In the normal direction, viscoelastic interaction is assumed [29]. The normal contact force is composed of two components; one is the elastic component caused by deformation or overlap, and the other is the viscous dissipation component caused by the energy dissipation of solid powders.

$$\overrightarrow{F_n} = \min(0, -E\xi^{3/2} - \frac{3}{2} A_n E \sqrt{\xi}\dot{\xi})\overrightarrow{e_n} \tag{11}$$

where $\xi = R_1 + R_2 - |\vec{r_1} - \vec{r_2}|$ is the compression of the particle radii $R_1$ and $R_2$ at positions $\vec{r_1}$ and $\vec{r_2}$, and $\vec{e_n} = (\vec{r_1} - \vec{r_2})/|\vec{r_1} - \vec{r_2}|$ is the normal unit vector. The elastic parameter $E$ is defined as:

$$E = \frac{2Y}{3(1-v^2)} \sqrt{R_1 R_2/(R_1 + R_2)} \qquad (12)$$

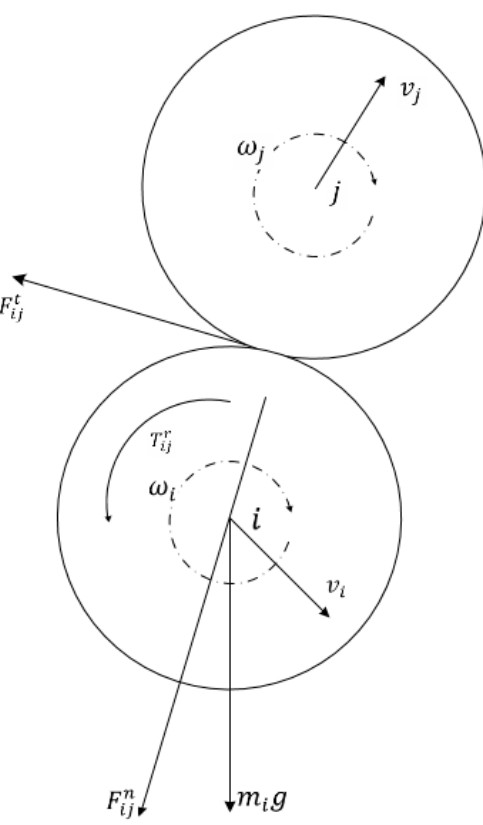

**Figure 4.** Schematic illustration of the forces acting on powder *i* in contact with powder *j*.

In the equation, $Y$ is Young's elastic modulus, $v$ is Poisson's ratio. $A_n$ is the dissipative parameter, which can be obtained by the method published in [30].

In the tangential direction, a modified Cundall–Strack model is applied.

$$\vec{F_t} = -\min\left[\mu|\vec{F_n}|, \int \frac{4G}{2-v} \sqrt{R_1 R_2/(R_1 + R_2)\xi}ds + A_t \sqrt{R_1 R_2/(R_1 + R_2)\xi}v_t\right]\vec{e_t} \qquad (13)$$

where $\mu$ is the Coulomb friction coefficient and $G$ is the shear modulus, which can be computed with the equation $G = Y/2(1+v)$. $v_t$ is the relatively tangential velocity, and $A_t$ is the tangential dissipative parameter, which can be calculated with the equation $A_t \approx A_n Y/(1-v^2)$.

Powder adhesion was computed with the Johnson-Kendall-Roberts (JKR) model; a detailed description of the JKR model can be found in [31]. For fine powders with diameters less than 100 μm, van der Waals forces are important in the packing densification of fine powders, so they cannot be negligible. Non-bonded van der Waals forces can be computed with the Hamaker expression with a cut-off distance of 1 μm.

$$\begin{aligned}
\vec{F_{vdw}} &= \frac{A_H \sqrt{\frac{R_1 R_2}{R_1 + R_2}}}{6D_{min}^2}\vec{e_n} & \xi > 0 \\[2mm]
\vec{F_{vdw}} &= \frac{A_H \sqrt{\frac{R_1 R_2}{R_1 + R_2}}}{6(\xi - D_{min})^2}\vec{e_n} & -1\,\mu m \leq \xi \leq 0
\end{aligned} \qquad (14)$$

$A_H$ is the Hamaker constant, which can be obtained with the equation $A_H = 24\pi D_{min}^2 \gamma$; $\gamma$ is the surface energy density, which is a characteristic of the powder material, $D_{min} = 0.165$ nm.

### 3.2. Establishment of Vibration Model

It is not possible to directly apply vibration to the particles in the software, so it was achieved by applying vibration to the bottom. For the application of vibration, force cannot be applied directly to the wall. The displacement equation can be converted into a velocity equation by applying velocity to the wall. In order to correspond to the experiment, the simple harmonic vibration method was adopted, and the displacement equation of its vibration is:

$$S = A\sin(2\pi ft + \varphi) \tag{15}$$

where $S$ is the displacement (mm), $A$ is the maximum amplitude (mm), $f$ is the vibration frequency (Hz), $t$ is the vibration time (s), and $\varphi$ is the primary phase. Through derivation, the velocity equation can be obtained.

$$v = 2\pi fA\cos(2\pi ft + \varphi) \tag{16}$$

## 4. Results and Discussion

### 4.1. Dependence of Packing Density on Powder Size Distribution

In order to validate the effect of the powder size distribution and powder size on packing densification, powders with different particle size distributions were used in the numerical simulation. The spherical powders with a single size (d = 30 μm, 300 μm, 30 μm is a typical powder size for additive manufacturing) and with a Gaussian distribution (the average diameter was 30 μm and 300 μm) of variance 0.2 were selected, and the material was a powder of 316 L stainless steel. Here, the vibration was not considered.

The rolling and sliding friction coefficients were 0.3 and 0.015, respectively [32], the damping ratio was 0.2 [32], and the material density was 7.98 g/cm$^3$. The container was cylindrical, with the height 30 mm, and the ratio of the inner diameter of the container to the radius of the powder was greater than or equal to 20, in order to eliminate the influence of the boundary size effect on the powder densification as much as possible.

At different depths of the container, the powder status was different. In order to measure the packing density at different depths of the container, Position A (5 mm, the distance from the bottom of the container), Position B (14 mm from the bottom), and Position C (23 mm from the bottom) were selected for monitoring.

For the simulation of the packing densification of powders with a mono-size and Gaussian distribution, the classic DEM model was used. However, for the simulation of the packing densification of fine powders with a mono-size and Gaussian distribution, the DEM model proposed in this paper was used. The simulation results are shown in Table 1; A, B, and C represent the depth of the powders in the container, E represents a mono-size distribution, G represents a Gaussian distribution, and 1 represents 30 μm powders with a single size and Gaussian distribution. AE represents the packing densification of the powders at a position 5 mm from the bottom of the container with an equal diameter distribution of 300 μm. BE represents the packing densification of the powders at a position 14 mm from the bottom of the container with an equal diameter distribution of 300 μm. CE represents the packing densification of the powders at a position 23 mm from the bottom of the container with an equal diameter distribution of 300 μm. AG represents the packing densification of the powders at a position 5 mm from the bottom of the container with Gaussian distribution (mean diameter is 300 μm). BG represents the packing densification of the powders at a position 14 mm from the bottom of the container with Gaussian distribution (mean diameter is 300 μm). CG represents the packing densification of the powders at a position 23 mm from the bottom of the container with Gaussian distribution (mean diameter is 300 μm). AE1 represents the packing densification of the powders

at a position 5 mm from the bottom of the container with an equal diameter distribution of 30 μm. BE1 represents the packing densification of the powders at a position 14 mm from the bottom of the container with an equal diameter distribution of 30 μm. CE1 represents the packing densification of the powders at a position 23 mm from the bottom of the container with an equal diameter distribution of 30 μm. AG1 represents the packing densification of the powders at a position 5 mm from the bottom of the container with Gaussian distribution (mean diameter is 30 μm). BG1 represents the packing densification of the powders at a position 14 mm from the bottom of the container with Gaussian distribution (mean diameter is 30 μm). CG1 represents the packing densification of the powders at a position 23 mm from the bottom of the container with Gaussian distribution (mean diameter is 30 μm). It can be seen from Figure 4 that the packing density of small powders is significantly different from that of large particles (>100 μm).

**Table 1.** Porosity measurement results at different depths of the container under different stiffness values.

| Powder Distribution | Powder Stiffness × e10 (*n*/m) | | | | | | |
|---|---|---|---|---|---|---|---|
| | 0.05 | 0.1 | 0.5 | 1 | 5 | 8 | 9 |
| AE | 0.175 | 0.255 | 0.348 | 0.351 | 0.364 | 0.367 | 0.368 |
| BE | 0.245 | 0.285 | 0.35 | 0.355 | 0.365 | 0.367 | 0.37 |
| CE | 0.34 | 0.335 | 0.365 | 0.365 | 0.372 | 0.371 | 0.371 |
| AG | 0.13 | 0.221 | 0.302 | 0.351 | 0.339 | 0.341 | 0.342 |
| BG | 0.18 | 0.256 | 0.318 | 0.355 | 0.339 | 0.342 | 0.344 |
| CG | 0.26 | 0.312 | 0.328 | 0.365 | 0.348 | 0.349 | 0.35 |
| AE1 | 0.32 | 0.37 | 0.42 | 0.446 | 0.455 | 0.458 | 0.461 |
| BE1 | 0.33 | 0.375 | 0.425 | 0.448 | 0.456 | 0.458 | 0.463 |
| CE1 | 0.34 | 0.379 | 0.429 | 0.451 | 0.458 | 0.459 | 0.464 |
| AG1 | 0.35 | 0.38 | 0.43 | 0.45 | 0.46 | 0.465 | 0.478 |
| BG1 | 0.355 | 0.386 | 0.433 | 0.455 | 0.468 | 0.469 | 0.48 |
| CG1 | 0.358 | 0.389 | 0.436 | 0.456 | 0.468 | 0.473 | 0.485 |

### 4.1.1. Simulation Result for Larger Powders

As is well known, the theoretical packing density of mono-size particles is 0.64. The calculated packing density of mono-size 300 μm powders was 0.63, which was consistent with the physical experimental results obtained by Scott, indicating that the numerical model is rational and feasible. The effect of the powder stiffness on the packing porosity for particles with a single size and particles with a normal size distribution is shown in Figure 4. It can be seen from the figure that with an increase in the particle stiffness coefficient, the particle porosity first increases obviously and then gradually stabilizes after the stiffness coefficient increases to $1 \times 10^{10}$ *n*/m. When the stiffness coefficient is less than $1 \times 10^{10}$ *n*/m, there is a large overlap between the particles in contact with each other. As the stiffness increases, the overlap decreases rapidly, so the porosity between the particles increases rapidly. When the stiffness coefficient exceeds $1 \times 10^{10}$ *n*/m, there is a small amount of overlap between the particles in contact with each other. As the stiffness increases, the overlap amount is not significantly reduced, so the porosity between the particles remains basically stable. The stable value of porosity at different depths was compared and analyzed. As the packing depth increases, the stable value of porosity gradually decreases because the gravity of the upper layer particles makes the strain between the lower layer particles relatively large; the amount of overlap at the contact points of the lower layer particles is increased, thereby reducing the porosity.

As is well known, same-size spheres in a closed-packing arrangement show 74% packing, whereas the same set of spheres with random closed packing exhibits about 63% space filling [25]. However, it can be observed from Figure 5 that the packing density is about 87% when the stiffness is equal to $5 \times 10^8$ *n*/m. As shown in Figure 5, the overlap phenomenon of the sphere accumulation at the bottom of the container is more obvious. When the rigidity of the sphere itself is small, the contact force inside the particles and the gravity of the upper particles work together to cause greater elastic

deformation between the particles in contact with each other. Therefore, the amount of overlap between particles is larger, and the overlap phenomenon is more obvious. Therefore, before powders are printed by additive manufacturing, the size of the powder cylinder and the amount of powder should be designed according to the rigidity of the specific material.

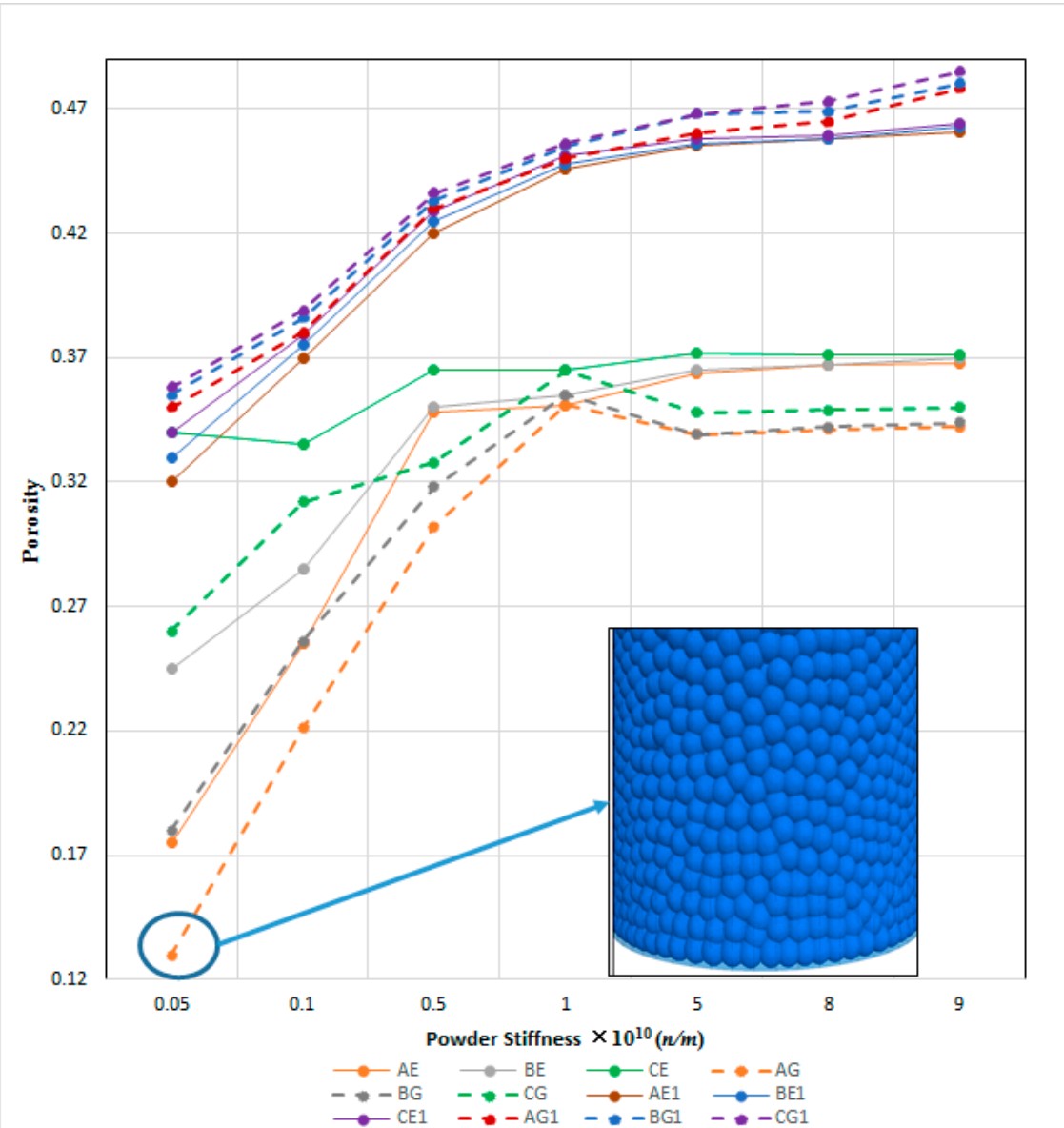

**Figure 5.** Porosity at different depths in container for mono-size particles and Gaussian size-distributed particles with different powder stiffnesses (E refers to the packing porosity for particles with a single size, and G represents the packing porosity for particles with a Gaussian size distribution).

### 4.1.2. Simulation Result for Fine Powders

In order to validate the DEM method for fine powders, comparison with the prediction from Yen [33] and Desmond [34] was implemented, which is shown in Figure 6. Yen conducted research on the packing density of mono-size powders. Desmond predicted the packing density of powders with a Gaussian distribution with a particle size greater than 100 μm under different variances. It can be seen from CASE 1 that the predicted packing densities with several methods for larger particle size powders are close to the theoretical value of 0.64. It can also be found from the prediction of CASE 2–CASE 6 that the method proposed in this paper is basically consistent with Yen's prediction. The packing density

of fine particles with a Gaussian distribution is lower than the case of mono-size. For the packing density of fine powders of 100 µm or less, the force between the powders (van der Waals forces and electrostatic attraction) plays a very important role.

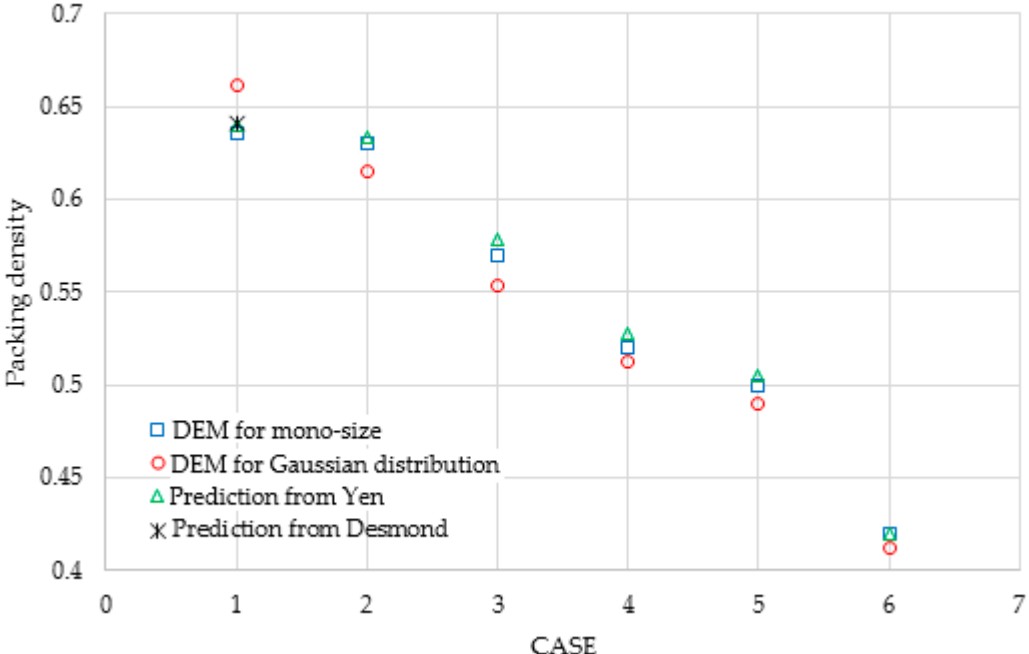

**Figure 6.** Comparison of packing density prediction by discrete element method (DEM) proposed in this paper with predictions from Yen and Desmond. (CASE 1: 300 µm powders [35], CASE 2: 100 µm powders without friction and Van der Waals forces, CASE 3: 100 µm powders with frictional coefficient 0.3 and without Van der Waals forces, CASE 4: 100 µm powders with frictional coefficient 0.3 and with Van der Waals forces, CASE 5: 100 µm powders with frictional coefficient 0.7 and with Van der Waals forces, and CASE 6: 50 µm powders with frictional coefficient 0.3 and with Van der Waals forces.)

For fine powders, the packing density is less than the theoretical value of 0.64. From Figure 5, we can observe that as the stiffness increases, the density of 30 µm powders tends to 0.53 (without van der Waals forces). Analyzing this phenomenon, it can be found that for fine particles, gravity no longer dominates, compared with large particles; powder agglomeration is more likely to occur, thereby reducing the packing density. Comparing the packing densities of fine powders with a mono-size distribution and Gaussian distribution, it can be found that the packing density with a Gaussian distribution is less than that with a mono-size distribution. This trend is obviously different from that of large powders. The reason could be that the powder size distribution is wider with a Gaussian distribution, and there are more small powder particles than in a mono-size distribution, which increases the possibility of agglomeration. If Van der Waals forces are considered, the packing density will be lower. In order to improve the packing density of the fine powder, vibration can be applied to effectively avoid the agglomeration of the powder.

*4.2. The Packing Density of Powder under Vibration*

In 2009, Li, Y.Q. et al. [36] conducted an experimental study on the dense packing of one-dimensional spherical particles under vibration conditions, and obtained a maximum packing density of 0.7131, which is close to the maximum theoretical value of 0.74. Through a stacking experiment for binary particles under vibration conditions, the maximum packing density of binary particles was obtained as 0.7522, which shows that binary particles can achieve higher-density dense packing under one-dimensional vibration conditions. In 2011, Li Chao et al. conducted an experimental study on the densification of binary-size one-dimensional and three-dimensional vibration

accumulation [17], and found that the maximum density of a binary-size ball under three-dimensional vibration was 0.8809.

Firstly, a three-dimensional vibration test was performed on stainless steel powder with a Gaussian distribution, an average particle diameter of 300 μm, and a standard deviation of 0.2. The experimental results at different amplitudes and different frequencies are shown in Figure 7. It can be seen from Figure 7 that the change trend of the powder packing density under different vibration conditions was consistent, but as the amplitude increased, the frequency for obtaining the maximum density decreased. This also shows that at larger amplitudes, lower frequencies are required to obtain the maximum density. The same conclusion can be obtained from the simulation result calculated with the simulation model established in Section 3.

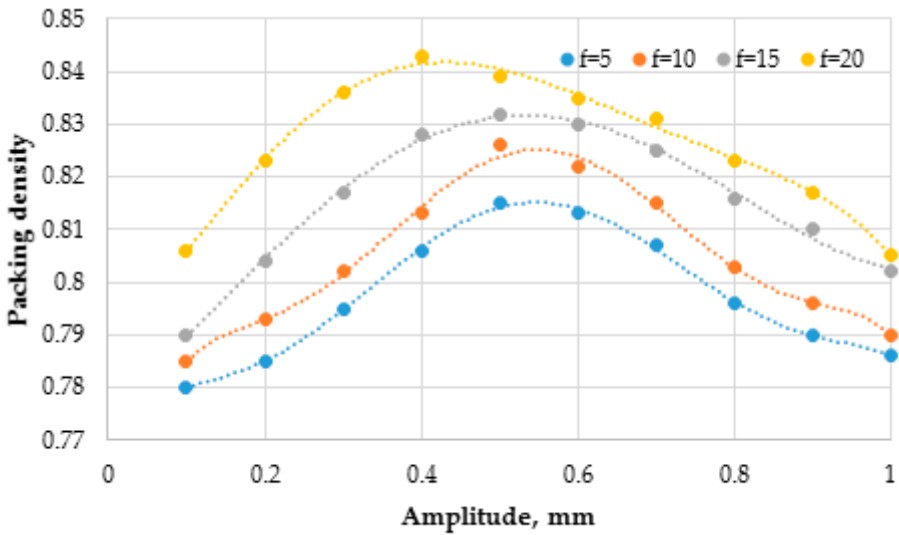

**Figure 7.** Effect of amplitude on packing density of powders with Gaussian distribution (average particle diameter of 300 μm and a standard deviation of 0.2) at different frequencies.

Secondly, experiments and numerical simulations were carried out on typical powders commonly used in 3D printing. In the previous numerical simulation results, it can be seen that the packing density of fine powder is lower without the application of vibration. In light of the fact that the powder agglomerates and reduces the density, the relationship between the three-dimensional vibration conditions and the packing density is discussed.

Figure 8 shows the experimental and numerical simulation results for the packing density of a powder with a Gaussian distribution with an average value of 30 μm, considering the three-dimensional vibration. It is obvious that the application of vibration can greatly increase the density. The results obtained by the numerical simulation analysis method proposed in this paper are in good agreement with the experimental results. It can be seen that at each fixed amplitude, in each size container, the packing density of the particles first increases with an increase in the vibration frequency, and after reaching the maximum value. When the maximum value is reached, it decreases as the frequency increases further. In addition to the influence of the vibration frequency on the densification of particle packing, amplitude also plays a vital role. According to the definition of vibration intensity, the magnitude of the amplitude directly affects the transmission of vibration energy in the particle packing, thereby affecting the rearrangement of particles in the process of vibration accumulation and densification. It can be seen that the influence of amplitude on particle packing densification is basically the same as the influence of vibration frequency; that is, for a given frequency, the particle packing density in each container first increases with an increase in amplitude. After reaching the maximum value, as the amplitude further increases, it decreases.

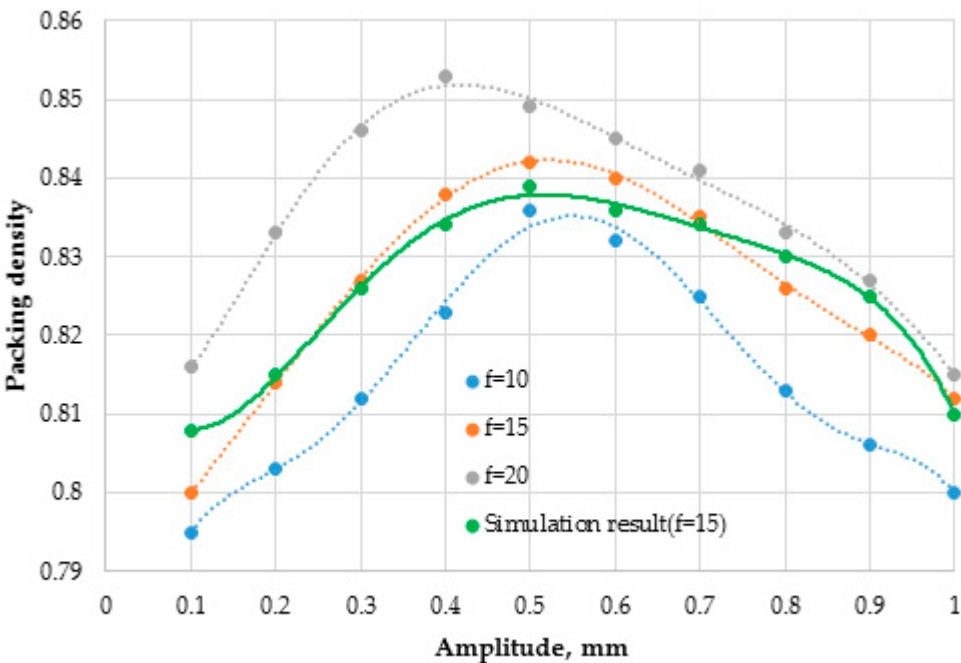

**Figure 8.** Effect of amplitude on packing density of powders with Gaussian distribution (average particle diameter of 30 μm) at different frequencies.

The influence of the vibration amplitude and frequency on the packing density was comprehensively analyzed, and the concept of vibration intensity was introduced. The vibration intensity is defined as:

$$\Gamma = A(2\pi f)^2/g \tag{17}$$

where $A$ is the vibration amplitude (m), $f$ is the vibration frequency (hz), and $g$ is gravitational acceleration.

Regardless of the influence of the vibration frequency or amplitude, according to the definition of vibration intensity, the factors are all attributed to the effect of the vibration intensity. Modifying the amplitude or vibration frequency can change the intensity of the vibration, thereby changing the energy input into the stacked powders. At each amplitude, the packing density of the powders first increases with an increase in the vibration intensity, and after reaching the maximum value, it decreases with a further increase in the vibration intensity. In the initial stage, the increase in vibration intensity increases the vibration energy transmitted between the stacked powders, which can effectively fill the porous structure formed in the initial pile. At the same time, it can release the lock between the powders and accelerate the particle partial rearrangement of the space, which can promote the process of densification. However, the experiment found that the vibration intensity cannot be too high, because a too-large intensity will cause the excessive excitation of the stacked powders, which will destroy the dense structure that has been formed, and reduce the packing density. In addition, from Figure 9, we also found that different packing densities are obtained at the same vibration intensity, and the same packing density can correspond to a different vibration intensity. Therefore, characterizing the packing densification that has formed should not only be based on a single vibration intensity, and the impact of the amplitude and vibration frequency should be considered separately.

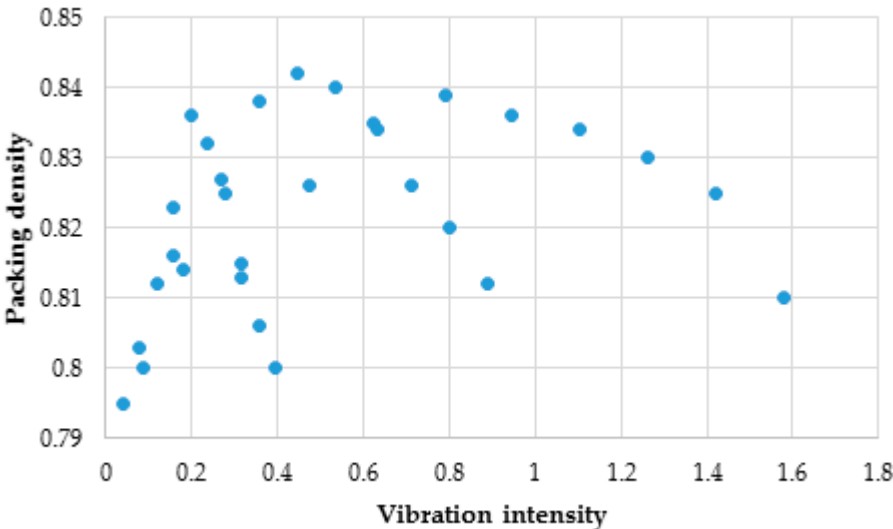

**Figure 9.** Packing density as a function of vibration intensity.

## 5. Conclusions

Aiming at the widely used powder spreading system and widely used typical powders in 3D printing, this paper used a combination of numerical simulation and experimental research to analyze the factors that affect the packing density of the powder in the powder container (these factors in turn affect the density of the finished part).

(1) The typical powder size of the powder used in 3D printing is less than 100 μm, which belongs to the category of fine powder. Taking into account the influence of van der Waals forces, a mathematical model suitable for fine powder was constructed on the basis of the classical discrete element model. At the same time, considering the influence of vibration on the packing density of powder, the displacement equation of simple harmonic vibration was converted into a velocity equation, and the function of the DEM mathematical model was expanded.

(2) A three-dimensional vibration experimental platform for powder was designed, and the influence of different vibration conditions on the packing density of powder was experimentally studied. The influence of the amplitude and vibration frequency on the powder packing density is the same; that is, it increases with an increase in amplitude or frequency, and then decreases with a further increase in amplitude or frequency after reaching the maximum. This shows that under each vibration condition, there is an optimal amplitude and frequency for achieving the densest packing of powders.

(3) The research results show that the compact structure formed by only relying on vibration intensity characterization is unreasonable. Therefore, it is necessary to combine the amplitude and frequency to analyze the factors that affect the density.

**Author Contributions:** Methodology, H.Y. and S.L.; validation, F.J.; investigation, Z.L.; writing—original draft preparation, S.L.; writing—review and editing, H.Y.; funding acquisition, H.Y. All authors have read and agreed to the published version of the manuscript.

**Funding:** This research was funded by the Science and Technology Foundation of the State Key Laboratory (Grant No. 9100419010) and the Fundamental Research Funds for the Central Universities (Grant No. 2017MS150).

**Conflicts of Interest:** The authors declare no conflict of interest.

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
