# Peer review of "Experimental and Numerical Study on the Packing Densification of Metal Powder with Gaussian Distribution"

_metals, doi:10.3390/met10111401_

Round 1
Reviewer 1 Report
REVIEW
on article
Huadong Yang, Shiguang Li, Zhen Li and Fengchao Ji
Summary. In this paper, the packing density of spherical powder particles with a Gaussian distribution is studied experimentally and numerically. Taking into account the van der Waals forces between powder particles, a discrete element model is proposed, suitable for fine metal powders for 3D printing. The authors designed and manufactured a vibration test bench and performed physical experiments on spherical metal powders with an approximately Gaussian distribution to obtain maximum density. The results obtained using the discrete numerical simulation method are compared with the experimental results. The authors analyzed the influence of the vibration amplitude and frequency on the powder packing density. It is shown that the density of the powder increases with an increase in amplitude or frequency, and then decreases with a further increase in amplitude or frequency after reaching the maximum.
However, the article has serious flaws and is written with errors.
- The aim of the study is the effect of vibration amplitude and frequency on the packing density of metal powder. The general trend of dependence is clear and obvious. It is not clear how this affects the quality of the final product? 0.5%, 1%, 5%? Moreover, this is precisely what is of interest.
- The article is framed inaccurately in violation of the requirements of the journal. Formulas are out of bounds on the page.
- Line 151-154. This is not a distribution function, but a distribution density function.
- Formula 4 for estimating the standard deviation is not written correctly.
- Formula 7 is also miswritten.
- Discrete elemental modelling is practically not described.
- The formulas written in Section 3.1 are not used further in article, algorithms and features of discrete element simulation are not revealed.
- Line 254. The equation for the speed is not written correctly.
- There is no comparison of the results obtained by the authors with the data of other researchers.
The article contains many errors and inaccuracies; its scientific value is debatable. I recommend rejecting the article.
Author Response
1.The aim of the study is the effect of vibration amplitude and frequency on the packing density of metal powder. The general trend of dependence is clear and obvious. It is not clear how this affects the quality of the final product? 0.5%, 1%, 5%? Moreover, this is precisely what is of interest.
Response1: Thank you very much for your suggestion. As we all know, the quality of the final part of additive manufacturing is affected by many influencing factors, such as powder quality, process parameters, gas flow distribution, and powder spreading quality. Researchers have done a lot of work on the influence of process parameters on part quality. Recent studies have found that the quality of the powder bed not only affects the density of the parts, but also affects the interaction process between the laser and the metal powder, which in turn affects the formation of the microstructure and the performance of the parts. For the quality of the powder bed, its influencing factors include the quality of the powder, the performance of the powder spreading system, and the parameters of the powder spreading. In addition, the initial powder packing state obviously also affects the spreading and distribution of the powder on the powder bed which will affect the quality of the final parts, especially for powders with original defects. Therefore, it is urgent to the explore how to improve the initial powder packing performance in the powder cylinder.
And also, please take a look at this paper “Alessandro Averardi, Corrado Cola, Steven Eric Zeltmann, Nikhil Gupta. Effect of particle size distribution on the packing of powder beds: A critical discussion relevant to additive manufacturing. Materials Today Communications, 2020,24,100964”, author point out “The effect of laser power, scanning speed, and powder bed layer thickness have been studied on parameters such as porosity in the printed part, surface roughness, and microstructure. Density of the packed bed is another parameter of critical importance as outlined by many studies but is among the least understood parameter in this field.” In the current research, the impact of the packing of particles in the powder cylinder on the quality of the final parts is ignored. Therefore, this article focuses on this unresolved problem.
Because there are many factors that affect the quality of the final part, it is difficult to accurately measure the percentage of any factor that affects the quality. Even for the process parameters that have been extensively studied, we can hardly say that its importance is 0.5%, 1%, or 5%.
Our point is that as long as it may affect the quality of the final product, no matter what its probability of occurrence, it is worth our academic research to improve the quality of printing.
2.The article is framed inaccurately in violation of the requirements of the journal. Formulas are out of bounds on the page.
Response 2: Thank you very much for your suggestion. We wrote the paper according to the template from official website of Metals journal, including the format of the formula. We checked it carefully and corrected the details.
3.Line 151-154. This is not a distribution function, but a distribution density function.
Response 3: Thank you very much for your suggestion. We corrected the error in the revised paper.
4.Formula 4 for estimating the standard deviation is not written correctly.
Response 4: Thank you very much for your suggestion. We corrected the error in the revised paper. The formula 4 in the revised paper is usually used in the field of powder engineering.
5.Formula 7 is also miswritten.
Response 5: Thank you very much for your suggestion. We corrected the error in the revised paper. The formula 7 in the revised paper is usually used in the field of powder engineering.
6.Discrete elemental modelling is practically not described.
Response 6: Thank you for your suggestion. The discrete element modeling method is a relatively mature method. The DEM modeling method used in this article is a supplement to the classic model, making it suitable for fine powder. For the classical model, we can refer to references 29, 30, and 31 in section 3.1, so we do not describe it in detail. Classical DEM algorithm is applicable and effective when dealing with large particles. However, we found that there are problems in handling fine particles. We have conducted an analysis on the packing density of fine powder particles and found that when analyzing the packing process of powders with an average diameter of 30μm, the influence of the van der Waals force, an important force, cannot be ignored.
7.The formulas written in Section 3.1 are not used further in article, algorithms and features of discrete element simulation are not revealed.
Response 7: According to the algorithm in 3.1, a program is written and performed to calculate the packing density. In the subsequent analysis, the simulation results are all based on the algorithm in section 3.1. The purpose of this article is to compare the simulation analysis methods and experimental results.
8.Line 254. The equation for the speed is not written correctly.
Response 8: We did not find the speed formula you mentioned in line 254. We guess that the formula you mentioned is the speed formula in line 241. We corrected the error in the revised paper.
9.There is no comparison of the results obtained by the authors with the data of other researchers.
Response 9: At present, we have not found any relevant research on the packing density of metal powder for 3D printing, so there is no comparison with other research results in this article. However, in order to verify the simulation results, we conducted experimental research.

Reviewer 2 Report
The authors improved the manuscript and satisfied most of my recommendations. I recommend hence it's publication in the present form.
Author Response
Thank you very much for your review comments.
Round 2
Reviewer 1 Report
- The authors answered my first remark in detail and enough. However, this is not reflected in the text of the article. Please make a comment in this article.
- My second note is not corrected, formulas are out of bounds of pages, formatted incorrectly, which makes it difficult to read.
- Formulas fixed but not formatted correctly.
- In response to my 6th comment, the authors recommend that I study papers 29, 30 and 31, instead of correcting the text of the article. Please explain the details of DEM modeling more clearly in the text of the article.
- Comparison of the data obtained by the authors with the results of other researchers is a necessary part of the article, which is published in a high-level journal. The results obtained by the authors are not eminently original. The trends and dependencies described in the article are obvious. Please add such a comparison.
Author Response
1.The authors answered my first remark in detail and enough. However, this is not reflected in the text of the article. Please make a comment in this article.
Response: Thank you very much for your suggestion. In the Introduction section, the scientific problem about this article is supplemented in line 38-54 . Thanks again.
2.My second note is not corrected, formulas are out of bounds of pages, formatted incorrectly, which makes it difficult to read.
Response: Thank you very much for your suggestion. We do not understand the meaning of the phrase "out of bounds of paper". In the revised manuscript, we adjusted the format of the formula. Is our understanding correct?
3. Formulas fixed but not formatted correctly.
Response: Thank you very much for your suggestion. We do not understand the meaning of the phrase "out of bounds of paper". In the revised manuscript, we adjusted the format of the formula. Is our understanding correct?
4. In response to my 6th comment, the authors recommend that I study papers 29, 30 and 31, instead of correcting the text of the article. Please explain the details of DEM modeling more clearly in the text of the article.
Response: Thank you very much for your suggestions. We have added a description of the solution process and basic principles of the DEM method in lines 223-257 of the revised manuscript.
5. Comparison of the data obtained by the authors with the results of other researchers is a necessary part of the article, which is published in a high-level journal. The results obtained by the authors are not eminently original. The trends and dependencies described in the article are obvious. Please add such a comparison.
Response: Thank you very much for your suggestion. In the revised manuscript (line 350-366), Figure 6 is added to compare the DEM method proposed in this paper with the Ye [34] and Desmond [35] forecasting methods.
Round 3
Reviewer 1 Report
REVIEW
on article
Huadong Yang, Shiguang Li, Zhen Li and Fengchao Ji
The article has been largely revised and looks much better now. The authors took into account all the comments, corrected the formulas, provided the necessary comments. The graphic part also looks much better, the figures are more informative and reflect the experimental part of the study.
The authors added a comparison of their results with those of other researchers, which will certainly attract the attention of readers.
I would like to wish the authors success and draw their attention to the more accurate design of the article, fulfilling all the necessary requirements of the editors. In future articles, try to enhance the theoretical part. Prepare the experimental part using modern statistical methods for processing the results.
I recommend the article for publication.
This manuscript is a resubmission of an earlier submission. The following is a list of the peer review reports and author responses from that submission.
Round 1
Reviewer 1 Report
Numerical Simulation on the Effect of Stiffness Coefficient and Container Size on Packing Density of Normally Distributed Metal Powder
The authors used DEM simulation to analyze the effect of powder material property, PSD, container size, and vibrations on packing density. In-house test rig was used to validate the DEM prediction. Overall, the manuscript does not present enough information by which another researcher could reproduce the results of this paper. Some terminologies and governing equations are not defined well or missed. The details are following below:
[1] The governing equations and terminologies used in the simulation should be well described. For instance, what kind of forces are assumed in the simulation? Adhesion force van der Waals force or friction force etc. are not mentioned or equated at all. Also, many the constancies and symbols are not defined in the manuscript. For example, on the line of 223, what is Fs and how do the authors define the shear force? On the line of 221, Ec is Young’s modulus not R. Also, R seems effective radius. Is it correct?
[2] The paper claims that the simulation is for AM process. However, the model was performed in a cylindrical container and particles simply dropped or created (It’s not described in the manuscript.). Powder spreading is the unique processing step distinguishing the packing in AM from the packing in traditional powder metallurgy. However, the current work does not model powder spreading at all, so it is hard to differentiate the packing process in this work from the one in conventional powder metallurgy.
[3] On the line of 140, the listed powder manufacturing method seems inappropriate to AM. In the AM, powder feedstock materials are typically fabricated via water automized, gas automized and PREP. The authors may refer to the website of https://www.carpenteradditive.com/technical-library/.
[4] Particle size of 300 μm (0.3 mm) is abnormally large for AM. The particle size is much bigger than typical laser beam diameter and similar to electron beam diameter. Please provide the reference or detailed spec from a measurement. For instance, accumulative frequency, D10, D50 and D90 etc.
[5] Figure 2 is not diagram.
[6] What is the input parameters used in the DEM simulation? What is rolling coefficient? What is value of cohesion force? How did the authors select material parameters? What is the rational behind for damping coefficient of 0.2?
[7] Please add color legend for Fig. 5.
[8] Please erase reference instruction.
[9] For Fig.4, the porosity becomes 0.13 (=packing density is about 87%). The model is done under 100% elastic assumption. How the packing density exceed the theoretical max packing density of 74% without deformation? Please provide a reference or reasonable explanation.
Reviewer 2 Report
In this paper, the authors investigated numerically (DEM simulations) the evolution of the packing density by considering particle size distribution, particle stiffness and packing size (here, container). This study is motivated by investigating powder packing density in the powder tank before spreading for the metal additive manufacturing. Three types of particle size distribution are considered: mono-size particles, random-distributed particles and normal-distributed particles. A separate experimental study for analyzing the vibration effects on packing density is also carried out without numerical comparison.
The paper is not well written and I do not recommend its acceptance in the present form. However, I encourage the authors to resubmit their work by considering the cited remarks.
- The numerical model is not well explained although I suppose that a classical DEM method is used. Please explain it more clearly, for example, by adding damping part, explaining why this definition of K_n and what is tangentiel force, etc. Moreover, in Introduction, the authors present the Molecular Dynamics (MD) method as a different method from the DEM for simulating particulate materials. Indeed, the classical DEM is an extension of the MD [Cundall and Strack (1979), Géotechnique]. It is also worth noting that there are other types of the DEM technique like the Contact Dynamics Method [Radjai and Richefeu (2009), Mechanics of Materials].
- In DEM simulations, the overlap between particles should be verified and a very large overlap is not authorized. Indeed, in DEM, we consider particles with small deformations. For example, when one consider the Hertzian contact law between particles, the DEM model is valid up to 10% of strain.
- In my point of view, the numerical studies presented in this paper are not original considering different studies in the domain of particulate (or granular) materials. Moreover, there are often no physical discussions to explain the observed results.
- As said before, an interesting experimental study is effectuated with no comparison with numerical results. This part should be omitted or enriched with a numerical comparison considering the paper subject.
- Although the manuscript is comprehensible, there are many typos in the text and academic english writing is not well. For exemple, the presentation of equations is not adequate. The authors should do proofreading before resubmitting the paper.
Reviewer 3 Report
REVIEW
on the article
Numerical Simulation on the Effect of Stiffness Coefficient and Container Size on Packing Density of Normally Distributed Metal Powder
Huadong Yang, Shiguang Li, Zhen Li and Fengchao Ji
Summary.
Usually, I start a review with the scientific problem that is reflected in the article. I didn't see the problem.
The relevance of the study is questionable.
The statistical part is considered at a primitive level without proof and discussion.
I think that the article has nothing to do with the Metals journal. Perhaps magazines considering bulk materials will be interested in this article.